

# The effect of the Buteyko breathing technique on asthma control and quality of life in children with asthma aged 7–12 years: a randomized controlled study

Hakan Çelik[*] and Emel Yuruk[*]

Faculty of Health Sciences, Cukurova University, ADANA, Turkey
[*] These authors contributed equally to this work.

## ABSTRACT

**Purpose**. This study was conducted to evaluate the effectiveness of Buteyko breathing technique in improving asthma control and quality of life in children aged 7–12 years. Given the increasing demand for alternative asthma treatments, this study aims to address the gap in evidence regarding the Buteyko breathing technique, thereby advancing clinical management of pediatric asthma.

**Method**. This study employed a randomized controlled design to evaluate the effectiveness of the Buteyko breathing technique in children with asthma. The study cohort consisted of 65 pediatric patients with asthma, who were receiving treatment at the Paediatric Allergy and Immunology Polyclinic. Thirty-three participants were assigned to the experimental group in which Buteyko breathing technique was applied in addition to standard asthma treatment and 32 participants were assigned to the control group in which standard asthma treatment was applied. The primary endpoints of this study were the patients' asthma control level and quality of life scores.

**Results**. Patients in the experimental group showed a statistically significant improvement in asthma control and quality of life with the Buteyko breathing technique ($p < 0.05$). A large effect size was observed for asthma control ($d = 3.54$) and quality of life ($d = 12.08$). The post-test asthma control scores of the experimental group were statistically significantly higher than the control group ($p < 0.05$; $d = 3.65$).

**Conclusion**. Children often have difficulty in asthma management due to difficulties in inhaler use. In this study, the Buteyko breathing technique was found to be effective in improving asthma control and quality of life in paediatric patients.

Corresponding author
Emel Yuruk, emelyuruk-bal@gmail.com

## INTRODUCTION

Asthma in childhood is characterized by chronic airway inflammation. Chronic inflammation of the airways causes symptoms such as recurrent morning and nighttime cough and chest tightness in children. These inflammations are explained by airway structural cells such as T lymphocytes, smooth muscle endothelial cells and eosinophils (*Global Initiative for Asthma, 2024*; *Türk Toraks Derneği, 2020*).

In children, gender, genetic predisposition, obesity, race, atopy and airway hyperreactivity are considered among the individual factors causing asthma. Infections caused by viruses and bacterial agents, allergens, diet, exposure to cigarette smoke and economic status in the society are environmental factors that cause asthma (*Global Initiative for Asthma, 2024*). When diagnosing pediatric asthma, different features should be considered according to the age of the patient (*Türk Toraks Derneği, 2020*). The diagnosis of asthma in early childhood (<5 years of age) starts with clinical evaluation. It is then determined with a detailed physical examination. The most common symptoms in this period are cough and wheezing (*Global Initiative for Asthma, 2024*; *Türk Toraks Derneği, 2020*). In children under five years of age, wheezing and crackles should be clearly identified, even though there is a potential risk that asthma may develop in the future (*Çelik, Soyer & Aydın, 2020*; *Global Initiative for Asthma, 2024*; *Türk Toraks Derneği, 2020*). An asthma attack in children is characterized by impairment in daily activities and nutrition, increased need for bronchodilator drugs, decreased exercise tolerance and respiratory rate. Chest tightness, attacks are accompanied by dyspnea, air hunger and wheezing symptoms (*Global Initiative for Asthma, 2024*; *Türk Toraks Derneği, 2020*).

Major studies such as the American Thoracic Society and the International Study of Asthma and Allergies in Childhood (ISAAC) show that approximately 4% to 23% of children in developed countries have asthma. There is an increase in childhood asthma and allergic diseases in developed countries. Environmental and individual factors are responsible for this increase in recent years (*Çelik, Soyer & Aydın, 2020*).

İn the study conducted by *Asher et al. (2006)*, childhood asthma was more common in the 6–7 age group than in the 13–14 age group. According to the study, which examined trends in asthma prevalence worldwide, the increase in the prevalence of asthma symptoms in children in participating centres was twice as high as in centres that saw a decrease (*Asher et al., 2020*). Two large multinational studies (European Community Respiratory Health Survey (ECRHS) and ISAAC) have mapped the prevalence of asthma in the world. However, it is thought that asthma is the cause of death in one out of 250 people who die in the world. With the adoption of modern lifestyle and increasing urbanization, this prevalence is expected to increase and it is predicted that 100 million more people will have asthma by 2025. It is estimated that 250,000 people may die in one year in the world due to asthma (*Gebresillasie et al., 2024*; *Global Initiative for Asthma, 2024*; *Türkiye Cumhuriyeti Sağlık Bakanlığı, 2018*). In Turkey, the prevalence of asthma varies between 6–15% in mass prevalence studies. Asthma is among the most important causes of morbidity and mortality in the world in terms of risk factors, precautions and treatments and constitutes a very serious social and economic burden (*Türkiye Cumhuriyeti Sağlık Bakanlığı, 2018*).

Asthma prevention includes all measures taken to prevent the occurrence of asthma in children or to alleviate the symptoms of the existing disease (*Global Initiative for Asthma, 2024*). These measures can be categorized under three main headings: drug therapy, education and control of environmental factors. While drug treatment controls symptoms such as shortness of breath caused by asthma, education provides information to the child and his/her family about the management of the disease. Control of environmental factors

involves eliminating or minimizing factors that trigger asthma such as diet, house dust mites, animal dander, mold, pollen, air pollution, cigarette smoke and infections (*Çelik, Soyer & Aydın, 2020*; *Türk Toraks Derneği, 2020*).

The frequency of treatments affects the child's ability to attend school, interact with friends, and reduces the time spent on play and leisure activities. In the case of chronic diseases, parents face challenges due to their inability to work and the increased financial burden on the family (*Can, 2024*; *Gürel & Şahiner, 2024*). Recurrent asthma attack symptoms may lead to difficulties in fulfilling daily activities (movement, nutrition, excretion, hygiene, sleep, *etc.*), intensification of emotional stress and increased dependence on others (*Gürel & Şahiner, 2024*; *Türk Toraks Derneği, 2022*). Disease self-management in children with asthma is of critical importance in terms of prevention of symptoms and reduction of asthma attacks (*Türk Toraks Derneği, 2022*; *Yaşar, Kanık & Yüksel, 2024*).

Nursing care for asthma patients should focus on monitoring the severity of asthma attacks and respiratory distress. Heart and respiratory rate and oxygen saturation should be evaluated. Oxygen should be given during attacks and prescribed oral or inhaled corticosteroids and bronchodilators should be administered. Adequate fluid intake is important to soften secretions and prevent fluid loss. If infection is present, antibiotics should be given on time. Nurses should also inform families about asthma triggers and support them in using effective respiratory methods by providing guidance on the correct use of medications (*Dardouri et al., 2021*; *Kaş & Yıldız, 2021*; *Kaş & Balcı, 2022*; *Gürel & Şahiner, 2024*; *Türk Toraks Derneği, 2022*; *Yaşar, Kanık & Yüksel, 2024*).

The demand for alternative asthma treatments is increasing and in this context the Buteyko breathing technique is also gaining more and more attention.

In recent years, respiratory methods have started to be used as one of the non-pharmacologic, alternative and/or complementary approaches in the management of many chronic diseases and conditions. The Buteyko breathing technique (BBT) was developed in Russia by Dr. Konstantin Buteyko in the 1950s. Dr. Buteyko evaluates normal respiration on the basis of carbon dioxide, the respiratory hold time used in the technique is defined as the control pause and this value shows the oxygenation and carbon dioxide tolerance of the body (*Buteyko Clinic International, 2023a*) BBT is a technique that is performed only through the nasal route and includes respiratory control and breath holding after respiration. The main aim of the technique is to prolong breathing in a controlled manner, including control pauses/ periods of breath holding to prolong this period over time (*Zeng et al., 2019*).

Breathing exercises are widely used in lung diseases. Many positive results have been reported in studies. In these studies, improvement in asthma symptoms, decrease in the use of $\beta$2- agonists, decrease in emergency room visits and hospitalizations, and increase in the rate of recovery from the hospital have been shown in patients with moderate or severe asthma who received inspiratory muscle training (*Buteyko Clinic International, 2023b*). Given the increasing demand for alternative asthma treatments, the BBT was developed to help meet the body's oxygen needs. Today, the method known as the BBT offers a drug-free treatment option for bronchial asthma. Patients who use the BBT instead of bronchodilators can control their symptoms (such as chest tightness, "wheezing"), and the

aim is to reduce the frequency and severity of symptoms. It has been reported that with the application of this method, asthmatic patients experience less mucus formation, less chest tightness, better sleep, feel more energetic and use less medication (*Hassan, Abusaad & Mohammed, 2022*; *Mohamed, Sayed & Abd-Elhamed, 2022*; *Vaishnav & Deepak, 2020*). This study aimed to evaluate the effects of BBT on asthma control and quality of life in asthmatic children aged 7–12 years.

## Hypotheses of the study

$H_{01}$: The mean change from baseline in asthma control score at the end of week 4 in the treatment arm is the same as in the control arm.

$H_{11}$: The mean change from baseline in asthma control score at the end of week 4 in the treatment arm is greater than in the control arm.

$H_{02}$: The mean change from baseline in quality of life score at the end of week 4 in the treatment arm is the same as in the control arm.

$H_{12}$: The mean change from baseline in quality of life score at the end of week 4 in the treatment arm is greater than in the control arm.

# METHOD

## Research design and sample

This study was planned as a single-center, parallel-group, pretest-posttest randomized controlled trial to determine the effect of the BBT on asthma control and quality of life in children aged 7–12 years receiving asthma treatment. The CONSORT-Outcomes 2022 Extension guideline was used in reporting the results of the study (*Butcher et al., 2022*).

The *post hoc* power of the study was calculated using the G*Power 3.1.9.7 programme (*Faul et al., 2007*). The calculation was based on the results of the independent sample $t$-test analysis. In the calculation, the asthma control pos$t$-test scale mean scores, which were statistically significantly different between the experimental and control group pos$t$-test scores, were used to determine the effect size. There were 33 participants in the experimental group and 32 participants in the control group. The mean posttest scale score of the experimental group was $18.21 \pm 3.15$ and the mean posttest scale score of the control group was $16.31 \pm 4.09$. In line with these data, the effect size was calculated as 3.65 ($d = 3.65$) and the power of the study was calculated as 99%. The minimum target for statistical power in nursing research is 80% (*Polit & Beck, 2017*). These findings show that the sample is sufficient. Families of children diagnosed with asthma provided informed consent and completed the pretests before being randomly assigned to either the intervention or control group. Block randomization was performed by a researcher who did not apply the BBT using a computer-assisted random sequence generator (http://www.random.org). Block size was six and equal allocation rate was applied. Blinding was used to select a sample from the population; the researcher applying the technique learned the randomization information after the pretest. Patients or researchers were not blinded, but data analysts were blinded to the study groups.

Children who had been diagnosed with asthma at least 6 months ago and were being treated according to the standard treatment protocol were included in the study. Children

with any disease that might prevent them from practicing the breathing technique (epilepsy, blood pressure imbalances, diabetes, cancer, hematological diseases, chest and/or heart pain, history of cardiovascular disease in the last 6 months, uncontrolled hyperthyroidism, brain tumor, visual and hearing impairment) were excluded from study.

The primary endpoints of this study were the change from baseline in the Asthma Control Test for Children (CISACT) score and the Quality of Life Scale in Asthmatic Children (QOLS) score at the end of week 4.

## Intervention

**Experimental group:** The child and his/her family were taught the BBT in addition to the standard treatment protocol. BBT is the most effective drug-free approach for the management of asthma and other respiratory health problems. The main aim of this technique is to balance the level of carbon dioxide in the body by reducing excessive breathing, thereby improving respiratory problems. According to Buteyko, many respiratory problems are actually caused by excessive breathing. Excessive breathing can lead to various health problems by lowering the level of carbon dioxide in the body. This technique includes various exercises to slow down the rate of breathing in and out and increase the duration of breath holding. Steps of this technique: the patient is seated in a comfortable position, the patient is made to breathe only through the nose, the patient is made to breathe superficially, the patient is made to hold the breath, the breath is released and finally the breath holding time is prolonged over time. A training video explaining BBT in a practical way is shown on the official website (*Buteyko Clinic International, 2023b*). The asthmatic child and his/her family were taught this technique practically. These training videos were then forwarded to the family so that the technique could be applied by the child and family at home. The participant continued to apply this technique at home for 10 min in the morning and evening for 4 weeks in addition to the treatment when he needed bronchodilator. The follow-up of this daily application was made over the phone with the parent of the child. At the end of 4 weeks, the participant who regularly applied the technique was given post-tests at the hospital (*Buteyko Clinic International, 2023b*).

**Control group:** The standard treatment protocol was applied to the child and his/her family in this group. The pre-test was applied to the children whose written and verbal consent was obtained from their families. The participant, who continued standard treatment and care at home, was examined in the hospital environment at the end of 4 weeks and post-tests were applied. After the research data were collected, the control group was shown BBT in practice. The research flow chart is given in Fig. 1.

## Study location

The study sample consisted of children aged 7–12 years with asthma who received outpatient treatment in Pediatric Allergy and Immunology Outpatient Clinic of Cukurova University Faculty of Medicine Hospital in Turkey between June and December 2023. Data were collected from the outpatient clinic of this hospital, which implemented a treatment protocol for paediatric patients with asthma. Within the scope of this protocol, drug treatment is planned for each age group, taking into account the severity and frequency

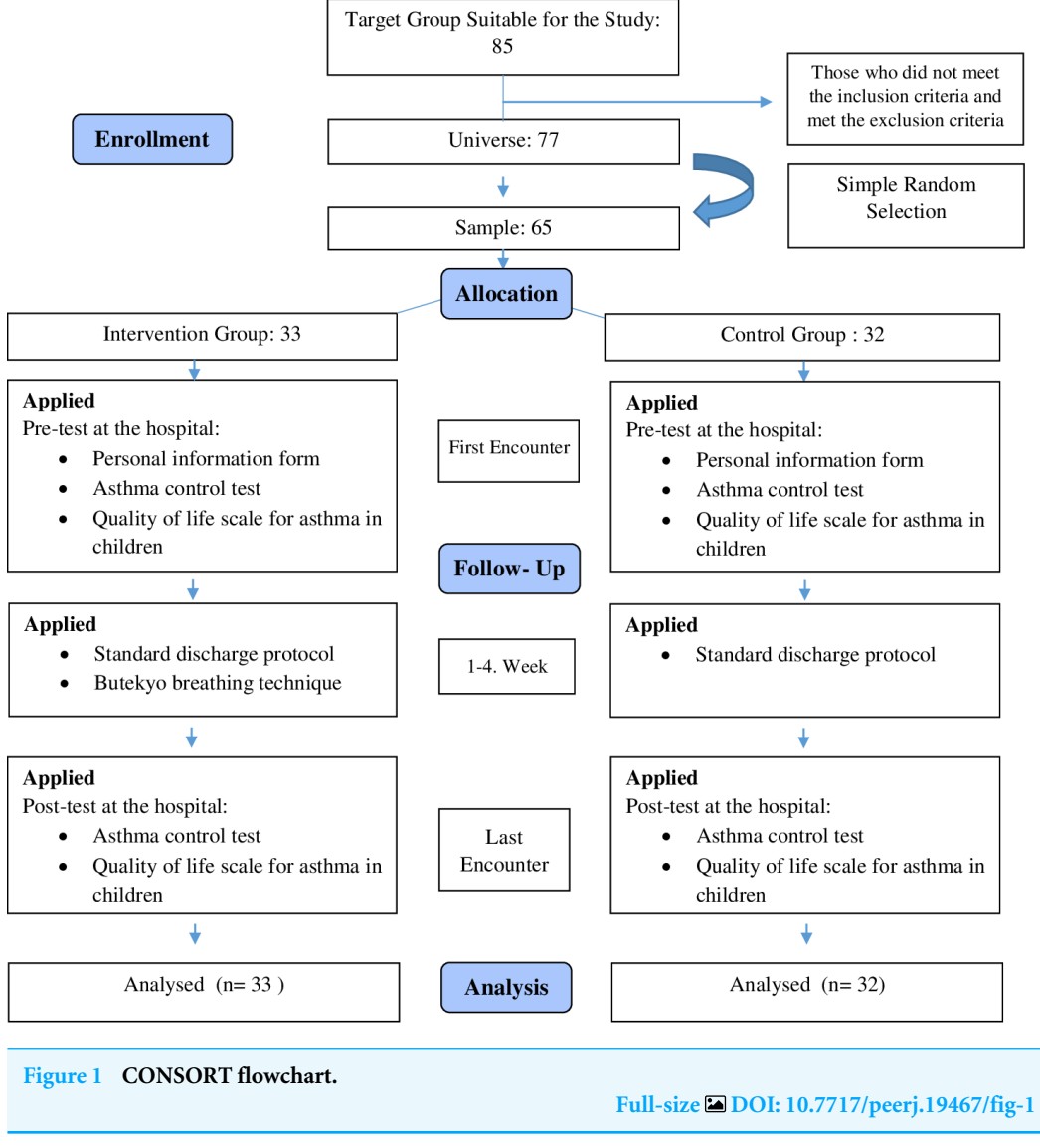

**Figure 1   CONSORT flowchart.**

of the disease, triggering factors and body mass index (body mass index (BMI), kg/m$^2$) measurements. In this direction, the parents of the paediatric patients who were examined and whose treatment was planned were contacted and invited to participate in the study, and the final examinations of the children whose follow-up was continued at home were collected at the hospital.

## Data collection tools
### Personal information form
The questionnaire form was formed by the researcher in line with the relevant literature information and included questions about the descriptive characteristics of the family (age, education, income status, *etc.*) (*Buteyko Clinic International, 2023a*; *Buteyko Clinic International, 2023b*; *Dardouri et al., 2021*; *Hassan, Abusaad & Mohammed, 2022*; *Kaş*

& *Yıldız, 2021*; *Kaş & Balcı, 2022*; *Mohamed, Sayed & Abd-Elhamed, 2022*; *Vaishnav & Deepak, 2020*; *Zeng et al., 2019*).

## Asthma Control Test for Children (CISACT)

CISACT is an important test consisting of five questions, which is the most widely used and easily understood by patients and their families, and which indicates whether asthma is in a good or poor prognosis.

It is a 5-item asthma control questionnaire that identifies children whose asthma is not adequately controlled. CISACT is scored on a scale of 1–5 for each question. Patients give each question a score between one and five. The total score of the five questions constitutes the test result. A total score of 25 is considered as complete control, 24–20 as partial control and ≤19 as not under control (*Asher et al., 2006*).

## Quality of Life Scale in Asthmatic Children (QOLS)

The Quality of Life Scale in Asthmatic Children (QOLS) was developed by *Juniper et al. (1996)* for children aged 7–17 years, and its Turkish reliability and validity study was conducted by *Yüksel et al. (2009)*. The QOLS consists of three sub-units, namely 'symptoms', 'activity restriction' and 'emotional functioning' and 23 questions. In the 10-question 'symptoms' sub-unit (4, 6, 8, 10, 12, 14, 16, 18, 20 and 23), shortness of breath, wheezing in the chest, cough, chest tightness and fatigue are questioned. In the 8-question 'emotional function' sub-unit (5, 7, 9, 11, 13, 15, 17 and 21), emotions such as sadness, resentment, fear, anxiety, anger, feeling different or excluded are questioned. In the 5-question 'activity limitation' sub-unit (1, 2, 3, 19 and 22), physical, social, school and sleep conditions are questioned. At the beginning of the study, the patient was asked to choose three of the 39 activities that he/she had the most difficulty in doing during the last 1 week.

Each question in the scale is equally weighted. The recorded scores were directly analyzed and the results were calculated both as total quality of life and as "average score per question" for each domain. Scoring ranges from 1–7 for each item. 1 corresponds to "always" or "extremely uncomfortable" and 7 corresponds to "never" or "not at all uncomfortable". The total quality of life score was calculated from the average score of all questions. A high score indicates a good quality of life.

## Implementation of data collection tools

Data were collected from children aged 7–12 years who were treated for asthma as outpatients at the Pediatric Allergy and Immunology Outpatient Clinic of Balcalı Hospital between 01/06/2023 and 30/12/2023. Data were collected from the children of families who met the research criteria and agreed to participate in the study using the data collection form. Before starting the study, the "Informed Voluntary Consent Form" was read to the patients and their verbal and written consent was obtained. Patients who underwent pre-testing were assigned to the experimental and control groups according to the randomization table. The last examinations of the children whose follow-up and interventions were continued at home were performed at the hospital and post-test data were collected.

## Data analysis

Statistical analyses were performed using SPSS (IBM SPSS Statistics 24) software. The normal distribution of the data was evaluated by Shapiro–Wilk and Kolmogorov–Smirnov tests. In the analyses, $p$ value $<0.05$ was accepted for statistically significant difference. In the reporting of research findings, effect sizes for statistically significant differences are also indicated. Various analysis techniques were used to calculate effect sizes (*Cohen, 1988*). In $t$-test results, the effect size was evaluated with Cohen's d value and the scaling is as follows: small (0.2), medium (0.5) and large (0.8 and above). Patient characteristics were summarized as frequencies and percentages for categorical variables, and as mean and standard deviation for continuous variables. Fisher's exact test or Pearson's chi-square test was used to compare categorical variables between the two groups, while a two-sample $t$-test was used to compare continuous variables between the two groups (*Pallant, 2020*).

## Ethical approval

Before starting the study, approval was obtained from the Cukurova University Clinical Research Ethics Committee with decision number 49 at meeting number 125, the participants were informed about the purpose of the study and written informed consent was obtained from the parents. No financial resources were used. The ClinicalTrials number of the study is NCT05793866.

## RESULTS

In the experimental group, 51.5% of the patients were female and 72.7% were primary school graduates. It was found that 72.7% of the patients lived in the city, 36.4% had a nuclear family structure, and 45.5% of them had a family income equivalent to expenses. It was found that 51.5% of the patients in the experimental group had asthma in their families, 48.5% of their mothers were primary school graduates and 57.6% of their fathers were high school graduates.

In the control group, 53.1% of the patients were female and 71.9% were primary school graduates. It was found that 75% of the patients lived in the city, 56.3% had a nuclear family structure, and 46.9% had a family income equivalent to expenses. It was found that 46.9% of the patients had a family history of asthma, 46.9% of their mothers were primary school graduates and 50% of their fathers were university graduates.

There was no statistically significant difference between the groups in terms of socio-demographic characteristics, asthma control and quality of life pretest mean scores ($p > 0.05$) (Table 1).

A statistically significant difference was found between the pretest and posttest asthma control mean scores of the participants in the experimental group ($p < 0.01$). When the mean scores were analysed, it was determined that the posttest asthma control score (18.21 $\pm$ 3.15) was higher than the pretest score (14.60 $\pm$ 3.29) and this difference had a large effect size ($d = 3.54$). On the other hand, there was no statistically significant difference between the pretest and posttest asthma control mean scores of the participants in the control group ($p > 0.05$).

**Table 1 Comparison of the descriptive characteristics of the patients in the experimental and control groups ($n = 65$).**

| Categorical variables | Experimental group ($n = 33$) Number (%) | Control group ($n = 32$) Number (%) | Test value | p value |
|---|---|---|---|---|
| **Gender** | | | | |
| Girl | 17 (51.5) | 17 (53.1) | .000 | 1.000[a] |
| Male | 16 (48.5) | 15 (46.9) | | |
| **Education status** | | | | |
| Primary School | 24 (72.7) | 23 (71.9) | .000 | 1.000[a] |
| Middle School | 9 (27.3) | 9 (28.1) | | |
| **Place of residence** | | | | |
| Rural | 9 (27.3) | 8 (25) | .000 | 1.000[a] |
| Urban | 24 (72.7) | 24 (75) | | |
| **Family structure** | | | | |
| Immediate family | 12 (36.4) | 18 (56.3) | 2.604 | .272[b] |
| Multigenerational household | 10 (30.3) | 7 (21.9) | | |
| Single-parent household | 11 (33.3) | 7 (21.9) | | |
| **Asthma in the family** | | | | |
| Yes | 17 (51.5) | 15 (46.9) | .140 | .708[b] |
| No | 16 (48.5) | 17 (53.1) | | |
| **Family income status** | | | | |
| Income less than expenditure | 12 (36.4) | 8 (25) | 1.385 | .500[b] |
| Income equals expenditure | 15 (45.5) | 15 (46.9) | | |
| Income more than expenditure | 6 (18.2) | 9 (28.1) | | |
| **Mother's education** | | | | |
| Primary School | 16 (48.5) | 7 (21.9) | 5.691 | .058[b] |
| High School | 8 (24.2) | 15 (46.9) | | |
| University | 9 (27.3) | 10 (31.3) | | |
| **Father's education** | | | | |
| Primary School | 3 (9.1) | 4 (12.5) | 2.649 | .267[c] |
| High School | 19 (57.6) | 12 (37.5) | | |
| University | 11 (33.3) | 16 (50) | | |
| **Continuous variables** | **Mean ± ss** | **Mean ± ss** | **t test** | **p value** |
| Age | 9.7 ± 1.5 | 9.4 ± 1.8 | .858 | .394[d] |
| Asthma year | 2.9 ± 2.3 | 2.9 ± 2.6 | .106 | .916[d] |
| Asthma control pretest mean scores | 14.6 ± 3.3 | 15.4 ± 3.5 | −.987 | .327[d] |
| Quality of life pretest mean scores | 103.4 ± 13.7 | 106.8 ± 15.9 | −.922 | .360[d] |

**Notes.**
Mean, Average; ss, Standard Deviation.
[a]Contunity correction.
[b]Pearson Chi-Square.
[c]Fisher's Exact Test.
[d]İndependent sample $t$ test.

A statistically significant difference was also found between the pre-test and post-test quality of life mean scores of the experimental group participants ($p < 0.01$). In terms of quality of life scores, it was determined that the posttest score (115.51 ± 13.78) was

**Table 2   Results regarding the comparison of the mean asthma control and quality of life scores of the experimental and control groups within and between groups.**

| Measurement | Pre test $\bar{X} \pm sd$ | Post test $\bar{X} \pm sd$ | Test Statistic | Cohen d 95% Cl |
|---|---|---|---|---|
| **Asthma control score** | | | | |
| Experimental | $14.60 \pm 3.29$ | $18.21 \pm 3.15$ | $-5.845$ 0.000[a] | Cohen $d = 3.54$ (1.434, .590) |
| Control | $15.43 \pm 3.49$ | $16.31 \pm 4.09$ | $-1.592$ 0.121[a] | |
| *Test statistic* | $-.987$ .327[b] | 2.096 0.040[b] | | |
| *Cohen d 95% Cl* | | Cohen $d = 3.65$ (.023, 1.013) | | |
| **Quality of life score** | | | | |
| Experimental | $103.39 \pm 13.68$ | $115.51 \pm 13.78$ | $-5.763$ 0.000[a] | Cohen $d = 12.08$ (1.418, .578) |
| Control | $106.78 \pm 15.89$ | $109.96 \pm 19.79$ | $-1.883$ 0.069[a] | |
| *Test Statistic* | $-.922$ .360[b] | $-5.845$ 0.197[b] | | |

Notes.

$\bar{X}$, Average; sd, Standard deviation.

[a]Paired sample $t$ test.

[b]İndependent sample $t$ test.

higher than the pre-test score ($103.39 \pm 13.68$) and this difference had a large effect size ($d = 12.08$). On the other hand, there was no statistically significant difference between the pre-test and post-test quality of life mean scores of the control group participants ($p > 0.05$) (Table 2).

When the posttest asthma control mean scores of the experimental and control groups were compared, a statistically significant difference was found between the groups ($p < 0.05$). Considering the posttest asthma control mean scores, it was determined that the mean score of the experimental group ($18.21 \pm 3.15$) was higher than the mean score of the control group ($16.31 \pm 4.09$) and this difference had a large effect size ($d = 3.65$). However, no statistically significant difference was found between the posttest quality of life mean scores of the experimental and control groups ($p > 0.05$) (Table 2).

# DISCUSSION

Our study shows that the socio-demographic characteristics of the experimental and control groups had similar characteristics at baseline and formed an appropriate basis for evaluating the effectiveness of the interventions ($p > 0.05$) (Table 1). The fact that there was no significant difference between the groups at the beginning of the study enabled us to measure the effect of the interventions more clearly.

The statistically significant increase in asthma control and quality of life scores of the experimental group indicates that BBT may be an effective non-pharmacologic treatment method ($p < 0.001$) (Table 2). In the study of *Mohamed, Sayed & Abd-Elhamed (2022)* comparing the effects of Buteyko and Pranayama breathing techniques on asthmatic

children, a decrease in the severity of asthma symptoms and improvement in quality of life related to disease control were observed in children who underwent the Buteyko technique. In the study of *Hassan, Abusaad & Mohammed (2022)* evaluating the effect of BBT on asthma severity control in school-age children, at the end of 4 weeks, the mean of the pos*t*-test of childhood asthma control was higher and statistically significant than the pre-test. The study supports the effectiveness of the BBT in improving respiratory outcomes and promoting asthma control in school-aged children with bronchial asthma. In additional findings, improvement in lung function test was observed (*Hassan, Abusaad & Mohammed, 2022*). Our study findings are consistent with previous studies and support the potential of Buteyko in asthma control. Similar studies in the literature show that the BBT is effective in alleviating symptoms, especially in patients with bronchial asthma, and reduces the use of bronchodilators (*Jena & Pradhan, 2020*). The positive effect of breathing exercises on alleviating symptoms in asthmatic patients is associated with relieving airway obstruction by reducing airway contraction. These results suggest that breathing exercises can be used as an effective treatment method in addition to pharmacologic methods used in asthma management (*Vaishnav & Deepak, 2020*).

The improvement in asthma control, especially in the experimental group, is associated with the ability of the BBT to reduce the need for bronchodilators and alleviate bronchial spasms (*Vagedes et al., 2021*). Studies have shown that this technique has the potential to reduce bronchial contractions, thereby reducing the frequency of asthma attacks (*Patil, Rani & Kumar, 2021*; *Vagedes et al., 2021*). In addition, breathing exercises such as the BBT are known to improve asthma management in children and improve overall quality of life. This technique, which focuses on respiratory control and the amount of air used during breathing, can significantly reduce the frequency and severity of asthma attacks while alleviating breathing difficulty and chest tightness in children (*Jena & Pradhan, 2020*; *Vagedes et al., 2021*; *Vaishnav & Deepak, 2020*).

Our study comparatively examined the effect of BBT on asthma control. The fact that the post test asthma control scores of the experimental group were significantly higher ($p < 0.05$) compared to the control group reveals the effectiveness of this technique (Table 2). In the literature, it is frequently emphasized that breathing exercises are beneficial in asthma management and relieve symptoms in patients with bronchial asthma (*Hepworth et al., 2019*; *Vagedes et al., 2021*). However, there was no statistically significant difference in quality of life scores between the groups ($p > 0.05$).This result suggests that although breathing exercises are effective in intra-group quality of life, improvement in inter-group quality of life may require longer-term treatment or a multifaceted approach.

In a study of *Valero-Moreno, Montoya-Castilla & Pérez-Marín (2023)* evaluating the quality of life in asthmatic patients aged between 12 and 16 years, the quality of life of adolescents was also affected by other factors such as the severity of asthma disease that they could not control, their psychological status (such as self-esteem, anxiety, depression) and family relationships. As in this study, signs and symptoms of asthma negatively affect the quality of life of adolescents with asthma. Again, *Agrawal et al. (2021)* examined the effect of severity of asthma on quality of life in school-age children and found that children

with asthma had a lower quality of life score ($p = 0.04$, $d = 0.53$) than children without asthma, and a decrease in asthma severity was associated with higher quality of life.

In the study conducted by *Lina et al. (2013)* in children with asthma, BBT was applied to children aged 7–11 years for 3 weeks. While there was no significant difference between the Asthma Control Test (ACT) results in the control group ($p = 0.177$), the experimental group showed a statistically significant improvement after the BBT was applied ($p = 0.002$). In addition, a significant improvement in quality of life was also observed in the experimental group at the end of the 4th week ($p = 0.002$) .

In the study of *Kavitha & Kalyani-Devi (2019)*, it was found that the BBT provided significant improvements (with pos*t*-test results) in the quality of life assessments of 80 school-age asthmatic children. This shows that BBT is effective in improving the quality of life of children with asthma.

Similarly, *Elnady et al. (2019)* examined the relationship between asthma control and quality of life in a study on Egyptian asthmatic children. Children with poor asthma control had a lower quality of life and this relationship was significant ($p < 0.005$). Children with poorly controlled asthma had a lower quality of life than children with well-controlled asthma.

Finally, *Eriksson et al. (2024)* found that patients who scored 20 points or more on the ACT had a higher quality of life. This is another study confirming the effect of asthma control on quality of life.

These studies suggest that BBTs have a significant ameliorative effect on asthma control, but their impact on quality of life may be more limited.

In conclusion, this study reveals that BBT plays an important role in improving asthma control and quality of life in children with asthma. Uncontrolled asthma can have serious negative effects on quality of life (*Abouelala et al., 2017*; *Banjari et al., 2018*). These findings are consistent with similar studies in the literature on the effects of the technique on asthma and emphasize the importance of holistic approaches in asthma treatment. In this context, it is understood that alternative breathing techniques such as BBT should be considered for an effective treatment.

## CONCLUSION

Asthma is a serious global health problem affecting all age groups. There are genetic, personal and environmental factors that cause asthma. Its prevalence is increasing in many countries, especially among children. Children find it more difficult to use inhalers than adults, leading to poor control of the disease. Correct use of inhalers is critical to prevent asthma symptoms and control the disease; misuse can increase the frequency of asthma attacks and reduce quality of life.

In conclusion, the BBT in children with asthma has positive effects on asthma control and quality of life. The regularity of breathing exercises plays an important role in controlling how much time is spent breathing. This technique provides benefits such as reducing bronchodilator feeding, moderating its consumption and ensuring overall health. It is also possible to improve both the physical and psychological outlook of children with asthma,

improving their quality of life. The combined evaluation of the BBT's contribution to the control of asthma symptoms and treatment processes strengthens the potential for its inclusion in the asthma management protocol.

## LIMITATIONS

In this study, the difficulties children faced in the process of learning and practicing the technique was an important factor limiting the effectiveness of the activity. In addition, the need for parental support limits the dissemination and sustainability of the implementation independently. The fact that the study was conducted in only one center limited the generalizability of the findings, reducing the possibility of assessing the possibility of achieving similar results in different geographical or socioeconomic contexts. These limitations can be taken into account in future research and contribute to the design of more comprehensive and generalizable studies.

## ACKNOWLEDGEMENTS

We would like to thank the staff of the Allergy and Immunology Unit of the hospital for their support during the data collection process and the parents of the children who participated in the study for their patience and contributions.

### Funding
The authors received no funding for this work.

### Competing Interests
The authors declare there are no competing interests.

### Author Contributions
- Hakan Çelik conceived and designed the experiments, performed the experiments, analyzed the data, prepared figures and/or tables, authored or reviewed drafts of the article, and approved the final draft.
- Emel Yuruk conceived and designed the experiments, performed the experiments, analyzed the data, prepared figures and/or tables, authored or reviewed drafts of the article, and approved the final draft.

### Human Ethics
The following information was supplied relating to ethical approvals (i.e., approving body and any reference numbers):

Before starting the study, approval was obtained from the Çukurova University Clinical Research Ethics Committee with decision number 49 at meeting number 125, the participants were informed about the purpose of the study and written informed consent was obtained from the parents.

## Data Availability

The dataset is available in the Supplemental File.

## Supplemental Information

Supplemental information for this article can be found online at http://dx.doi.org/10.7717/peerj.19467#supplemental-information.

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
