# Peer review of "The effect of the Buteyko breathing technique on asthma control and quality of life in children with asthma aged 7–12 years: a randomized controlled study"

_PeerJ, doi:10.7717/peerj.19467_

## Round 0.1 · original submission · Major Revisions

Please respond to the comments from the reviewers, in full

Reviewer 1 ·

Basic reporting

Clear professional English used throughout the manuscript.

Literature references is relevant and updated.

Professional article structure, figures, tables. Raw data shared are clear and understandable

Self-contained with relevant results to hypotheses.

Experimental design

The methods section was clear and well-written, with a detailed explanation of the research process. It was easy to follow and transparent, making the study replicable.

No comments

Validity of the findings

The findings effectively addressed the study's hypotheses, and the results supported the original objectives.

Conclusions are well stated, linked to original research question & limited to supporting results.

Additional comments

No comments

Reviewer 2 ·

Basic reporting

o The organization of abstract section can be improved
 The “purpose” section should include the rationale for conducting the clinical trial
 “Method” section should describe the basic study design, study treatment, key characteristics of patient population and primary/secondary endpoints/objectives. The details about the name of the hospital, time and whether the study is approved by ethics committee are not necessary in the abstract
 “Results” section presents an error regarding statistical significance: statistically significance means p value <0.05 rather than >0.05 (if significance level is set to 0.05). And some statements are contradictory: the author mentioned that “Patients in the experimental group showed a statistically significant improvement in asthma control and quality of life with Buteyko Breathing Technique”, however, they later stated that “no significant difference was found between the groups in quality-of-life scores”
 Conclusion section: grammar issue: “We suggest that Buteyko Breathing Technique plays an important role.”
o The organization of “Method” section can be improved
 At the beginning of this section, overall study design should be mentioned, including patient allocation scheme, what treatment will be used, what is the key characteristics of patient population, treatment information and duration, study endpoints including primary and secondary endpoints.
 Treatment information presented in Line 210-235 is the key part in the method section, and is suggested to be included in the basic study design at the beginning rather than at the end of the section
 Line 157-164 regarding randomization should be in research design section
 “Data analysis” section should be polished to include statistical methods used in this analysis, see comments in experimental design.
o In general, the language of this manuscript can be improved, some examples of grammar issue and typos:
 Duplicate words: Line 43: “obesity, obesity”, line 51-52 “wheezing and wheezing”
 Line 63: “in the study conducted Acher et al. (2006)”, should be “conducted by”
 Grammar issue: Line 64-65: “Increases in the prevalence of childhood asthma are twice as common as decreases…”, Line 86-87: “The frequency of treatments causes the child to go to school, communicate with friends and spend less time in play and entertainment.”
 Line 119: “Buteyko Inhalation Technique”, please clarify its difference with BBT
 Line 157: “Families of children diagnosed with asthma…”
 Line 203: missing hospital name: “of XXX hospital”
 Line 240-241: “For normally distributed data, t-test was used and statistical significance level was accepted as p<0.05.”. It is not the statistical significance level that’s accepted. It should be rephrased to: p value <0.05 is considered statistically significant.
 In table 2 and table 3, there should be a comma to separate lower and upper bound of 95% confidence interval

Experimental design

o Definitions of endpoints are missing
 The definition of the primary endpoint should be added. Based on this manuscript the authors are interested in evaluating the effect of the buteyko breathing technique on asthma control and quality of life. So the primary endpoints can be change from baseline in Asthma Control Test for Children (CISACT) score and Quality of Life Scale in Asthmatic Children (QOLS) score at end of week 4.
 And given that there are two primary endpoints, it should be noted in the manuscript whether the sample size will be calculated based on only one of primary endpoints or both
o Sample size calculation is not clear:
 From statistics perspective, to calculate sample size needed to achieve a desired power while controlling the type I error rate, we need to define the null hypothesis and alternative hypothesis based on primary endpoint. In the introduction section line 128-132, the authors defined two hypotheses of the study H1 and H2, from the statistical perspective, these are alternative hypotheses, and the authors should also define the null hypotheses. For example, the null hypothesis H0 associated with Asthma control score is: the mean change from baseline in Asthma control score at end of week 4 in treatment arm is the same control arm. The alternative hypothesis H1 is: the mean change from baseline in Asthma control score at end of week 4 in treatment arm is greater than control arm.
And it looks that both primary endpoints were statistically tested. If this is the case, sample size calculation should consider multiplicity issue.
 And it is not clear how “effect size” and “upper limit of the large effect size” are defined.
 Additionally, Line 154: “Alpha value was accepted as 0.05 and power as 0.90”, it should be rephrased to “Statistical significance level is set to 5% and desired power is 90%”.
o Statistical methods are not well-defined in line 237 Data analysis section:
 From table 1, patient characteristics are compared using two sample t test for continuous variables and Fisher’s exact test or Chi-square test for categorical variables. Such analysis method should be described in “Data analysis” section.
 From table 3, two sample t test was used to compare post-treatment asthma control test score and quality of life score between control and treatment arm. Such method should be described in “Data analysis” section.
 In Table 2, Paired-sample t test should have been used to compare pre- and post- treatment asthma control test score and quality of life score, instead of using two sample t test to compare them. This is because pre- and post- treatment scores are not independent samples and two sample t test is for comparing two samples when they are independent to each other. Such method should be described in “Data analysis” section.

Validity of the findings

o In table 2, Comparing pre- and post- treatment scores should be based on paired sample t test instead of two-sample t test

Reviewer 3 ·

Basic reporting

In the introduction, it is clearly stated that unmet needs for the asthma control therapy for kids and stated different kids may have different reason triggers asthma. In the proposed randomized study, there is a lack of statement for randomization stratification factors and how this study estimate the treatment effect while considering there are a lot of confounding effects, such as reasons that trigger asthma (food, environmental, or virus, etc), kids weight, height, health conditions, etc.

Experimental design

The experimental design states the sample size is 65, while it is not clearly state randomization strategy of the study. Also, this is a single center study, the authors may want to add justifications on this study to confirm this study would be representative for the whole population. Also, as these study samples are all asthma patients, who may need everyday treatment to control the disease, where different kids may have different drug or dose levels, the author may want to compare this confounding effect in the study.

Validity of the findings

Based on data in Table 1, it seems like the two treatment groups are balanced in the listed baseline characteristics. However, there are some important factors are missing, such as kids weight, height, BMI, health condition, time from last asthma symptoms, triggers for asthma, etc. These factors will also impact on the final analysis and its interpretation.

---

## Round 0.2 · Minor Revisions

Please revise your manuscript according to the reviewer's comments.
Yours,
Yoshi
Prof. Yoshinori Marunaka, M.D., Ph.D.

Reviewer 2 ·

Basic reporting

See additional comments

Experimental design

See additional comments

Validity of the findings

See additional comments

Additional comments

The authors have responded and addressed most of the comments in the first round of review, except for the following:
Line 45: what is “ing”? typo.
Line 57-59 it is still unclear the meaning of “Increases in the prevalence of childhood asthma are observed twice as often as decreases”, and this was not seen in the literature cited.

Line 262 to 264: please rephrase the sentence to: patient characteristics were summarized by number and percentage for categorical variables, and mean and standard deviation for continuous variables, Fisher’s exact test/Pearson chi-square test were used to compare categorical variables between two groups and two-sample t test was used to compare continuous variables between two groups.

Please move line 296 to 297 because it is intended for the quality of life outcome, so as to not confuse with the analysis for asthma control score.
İn table 2, only the post test asthma control score and quality of life score are compared between two arms, however, the primary endpoints are change from baseline at 4th week for asthma control score/quality of life score. Please also present the test for change from baseline.

Reviewer 3 ·

Basic reporting

The author addressed all my comments. No further comments from me.

Experimental design

The author addressed all my comments. No further comments from me.

Validity of the findings

The author addressed all my comments. No further comments from me.

---

## Round 0.3 · accepted · Accept

Congratulation!
Yours,
Yoshi
Prof. Yoshinori Marunaka, M.D., Ph.D.

Reviewer 2 ·

Basic reporting

No additional comments as the authors have addressed the previous comments.

Experimental design

No additional comments as the authors have addressed the previous comments.

Validity of the findings

No additional comments as the authors have addressed the previous comments.